

**Particle number concentrations and size distributions in the stratosphere: Implications of**
**nucleation mechanisms and particle microphysics**
Fangqun Yu[1], Gan Luo[1], Arshad Arjunan Nair[1], Sebastian Eastham[2,3], Christina J. Williamson[4,
5, a], Agnieszka Kupc[5, 6], and Charles A. Brock[5]
[1] Atmospheric Sciences Research Center, University at Albany, Albany, New York, US
[2] Laboratory for Aviation and the Environment, Department of Aeronautics and Astronautics,
Massachusetts Institute of Technology, Cambridge, MA 02139, USA
[3] Joint Program on the Science and Policy of Global Change, Massachusetts Institute of
Technology, Cambridge, MA 02139, USA
[4] Cooperative Institute for Research in Environmental Sciences, University of Colorado,
Boulder, CO 80309, USA
[5] Chemical Sciences Laboratory, National Oceanic and Atmospheric Administration, Boulder,
CO 80305, USA
[6] Faculty of Physics, Aerosol Physics and Environmental Physics, University of Vienna, 1090
Vienna, Austria
[a] now at: Climate Research Programme, Finnish Meteorological Institute, 00101 Helsinki,
Finland and Institute for Atmospheric and Earth System Research/Physics, Faculty of Science,
University of Helsinki, 00014 Helsinki, Finland.
Correspondence to: F. Yu (fyu@albany.edu)
**Abstract**. While formation and growth of particles in the troposphere have been extensively
studied in the past two decades, very limited efforts have been devoted to understanding these in
the stratosphere. Here we use both Cosmics Leaving OUtdoor Droplets (CLOUD) laboratory
measurements taken under very low temperatures (205–223K) and Atmospheric Tomography
Mission (ATom) in-situ observations of particle number size distributions (PNSD) down to 3 nm
to constrain nucleation mechanisms and to evaluate model simulated particle size distributions in
the lowermost stratosphere (LMS). We show that the binary homogenous nucleation (BHN)
scheme used in most of the existing stratospheric aerosol injection (a proposed method of solar
radiation modification) modeling studies overpredict the nucleation rates by 3–4 orders of



magnitude (when compared to CLOUD data) and particle number concentrations in the
background LMS by a factor ~2–4 (when compared to ATom data). Based on a recently developed
kinetic nucleation model, which gives rates of both ion-mediated nucleation (IMN) and BHN at
low temperatures in good agreement with CLOUD measurements, both BHN and IMN occur in
the stratosphere. However, IMN rates are generally more than one order of magnitude higher than
BHN rates and thus dominate nucleation in the background stratosphere. In the Southern
Hemisphere (SH) LMS with minimum influence of anthropogenic emissions, our analysis shows
that ATom measured PNSDs generally have four apparent modes. The model captures reasonably
well the two modes (Aitken mode and the first accumulation mode) with the highest number
concentrations and the size-dependent standard deviations. However, the model misses an apparent
second accumulation mode peaking around 300–400 nm, which is in the size range important for
aerosol direct radiative forcing. The bi-mode structure of accumulation mode particles has also
been observed in the stratosphere well above tropopause and in the volcano-perturbed stratosphere.
We suggest that this bi-mode structure may be caused by the effect of charges on coagulation and
growth, which is not yet considered in any existing models and may be important in the
stratosphere due to high ionization rates and long lifetime of aerosols. Considering the importance
of accurate PNSDs for projecting realistic radiation forcing response to stratospheric aerosol
injection (SAI), it is essential to understand and incorporate such potentially important processes
in SAI model simulations.







## 1. Introduction

Solar radiation modification (also known as solar geoengineering) approaches are being developed in response to the climate crisis (IPCC, 2021). They would temporarily offset climate change by reducing incoming sunlight, augmenting (currently inadequate) mitigation efforts and buying time to reduce atmospheric levels of $CO_2$, which is the root cause of the climate crisis. A recent report by the National Academies of Sciences, Engineering and Medicine (NASEM) emphasizes the urgent need to have a comprehensive understanding of the feasibility and potential risks/benefits of solar climate intervention approaches (NASEM, 2021). Stratospheric aerosol injection (SAI) has demonstrated the most promise as proximately engineerable (Shepherd et al., 2009; Lockley et al., 2020; IPCC, 2021) and has been extensively studied using models (e.g., GeoMIP: Kravitz et al., 2011; GLENS: Mills et al., 2017; Richter et al., 2022). The NASEM report (NASEM, 2021) pointed out that "the overall magnitude and spatial distribution of the forcing produced by SAI depends strongly on the aerosol size distribution" and "One of the research priorities for SAI is thus to address critical gaps in knowledge about the evolution of the aerosol particle size distribution". In the stratosphere, sulfate aerosols are formed by nucleation, followed by condensational growth and coagulation, and lost by evaporation in the upper stratosphere and downward sedimentation into the troposphere (Turco et al., 1982). New particle formation (NPF) (or nucleation) affects not only the number abundance but also the size distributions of stratospheric particles (e.g., Brock et al., 1995; Lee et al., 2003). There is increasing evidence (Weisenstein et al., 2022, Laakso et al., 2022) that a careful treatment of microphysical processes is necessary for projecting realistic radiative forcing response to SAI.

The process of NPF under tropospheric conditions has been extensively explored over the last two decades through laboratory and field measurements, theoretical studies, and numerical simulations (e.g., Yu and Turco, 2000; Vehkamäki et al., 2002; Kulmala et al., 2004; Kirkby et al., 2011; Dawson et al., 2012; Zhang et al., 2012; Kürten et al., 2016; Yu et al., 2018; Kerminen et al., 2018; Lee et al., 2019). Although some of the advances in our understanding of nucleation gained in the last two decades can be applied to stratospheric conditions, focused studies specifically examining the mechanisms of NPF under stratospheric conditions are quite limited. Indeed, the $H_2SO_4$–$H_2O$ binary homogenous nucleation (BHN) parameterization developed two decades ago by Vehkamäki et al. (2002) (named BHN_V2002 thereafter) has been used in most of SAI modeling studies when nucleation mechanism is considered (e.g., Weisenstein et al., 2022, Laakso et al., 2022). To our knowledge, the performance of this widely used BHN_V2002 under stratospheric conditions has not been carefully examined, probably due to the lack of suitable in situ measurements of freshly nucleated particles in the stratosphere for constraining the scheme. In this regard, particle size distributions down to 3 nm measured in-situ during the NASA Atmospheric Tomography Mission (ATom) in the lowermost stratosphere (LMS) of both SH and NH in four different seasons (Williamson et al., 2019, 2021; Kupc et al., 2020; Brock et al., 2021) provide much-needed data to constrain our understanding of the nucleation and particle microphysics in the stratosphere. In addition, well-controlled CLOUD experiments taken under low temperature (within the range of stratosphere) can also be used to assess the performance of nucleation schemes under stratospheric conditions. Another important issue related to stratospheric particles is the role of ionization in nucleation. It is well established that nucleation of $H_2SO_4$–$H_2O$ on ions is favored over homogenous nucleation (Hamill et al., 1982; Yu and Turco, 2000; Lovejoy et al., 2004; Kirkby et al. 2011; Yu et al., 2018) but the role of ionization in NPF



in the stratosphere has not been considered in any previous SAI studies (to our knowledge) in spite
of the very high ionization rates in the stratosphere.
In this study, we use both CLOUD laboratory measurements taken under very low
stratospheric temperatures and ATom PNSD measurements in LMS to constrain nucleation
mechanisms and model simulated particle size distributions. For 3-D simulation of size-resolved
stratospheric aerosols, we use the GEOS-Chem with the unified tropospheric-stratospheric
chemistry-transport model with the size-resolved advanced particle microphysics (APM) package.

**2. Model and data**
**2.1 GEOS-Chem/APM**
The GEOS-Chem model is a global 3-D model of atmospheric composition (e.g., Bey et al.,
2001) and is continuously being improved (e.g., Luo et al., 2020; Holmes et al., 2019; Keller et al.,
2014; Murray et al., 2012; Pye and Seinfeld, 2010; van Donkelaar et al., 2008; Evans and Jacob,
2005; Martin et al., 2003). The GEOS-Chem tropospheric–stratospheric unified chemistry
extension (UCX; Eastham et al., 2014), now the standard GEOS-Chem configuration, implements
stratospheric chemistry, calculation of J-values for shorter wavelengths, and improved modeling
of high-altitude aerosols. Extension of the chemistry mechanism to include reactions relevant to
the stratosphere enables the capturing of stratospheric responses and troposphere–stratosphere
coupling. UCX adds 28 species and 104 kinetic reactions, including 8 heterogeneous reactions,
along with 34 photolytic decompositions. Atomic oxygen [both $O(^3P)$ and $O(^1D)$] is explicitly
modeled; although also of short lifetime in the stratosphere, these species are important in correctly
modeling stratospheric chemistry. Photochemistry is extended up to the stratopause to high-energy
photons (177 nm) using the Fast-JX model, which includes cross-section data for many species
relevant to the troposphere and stratosphere. Photolysis rates respond to changes in the
stratospheric ozone layer. Additional heterogeneous reactions (Kirner et al., 2011, Rotman et al.,
2001, Shi et al., 2001) are included to capture seasonal ozone depletion. $H_2O$ is treated as a
chemically-active advected tracer within the stratosphere. These permit chemical feedbacks
between stratospheric ozone and aerosols and tropospheric photochemistry. The improved GEOS-
Chem with coupled stratospheric–tropospheric responses has been evaluated with sonde and
satellite measurements of $O_3$, $HNO_3$, $H_2O$, $HCl$, $ClO$, $NO_2$ and stratospheric intrusions (Eastham
et al., 2014; Gronoff et al., 2021; Knowland et al., 2022). Yu and Luo (2009) incorporated a size-
resolved (sectional) APM package into GEOS-Chem, henceforth referred to as GC-APM. The
APM separates secondary particles from primary particles, uses 40 bins to represent secondary
particles with high size resolution for the size range important for the growth of nucleated particles
to accumulation mode sizes, and contains options to calculate nucleation rates based on different
nucleation schemes. APM is fully coupled with GEOS-Chem in both the troposphere and
stratosphere, and is employed for the present study.
In the present study we have carried out GEOS-Chem-UCX/APM global simulations from
01/2015 to 05/2018, with the first 17 months as spin-up and the remaining period covering ATom
1-4 periods (06/2016–05/2018). The horizontal resolution is $4^o \times 5^o$ and there are 72 vertical layers.
Emissions from different sources, regions, and species are computed via the Harvard-NASA
Emissions Component (HEMCO) on a user-defined grid (Keller et al., 2014). Historical global
anthropogenic emissions are based on the Community Emissions Data System (CEDS) inventory



(Hoesly et al., 2018). Regional anthropogenic emissions over the United States, Canada, Europe,
and East Asia are replaced by regional emission inventories of the National Emissions Inventory
(NEI, https://www.epa.gov/air-emissions-inventories/2017-national-emissions-inventory-nei-
data), the Air Pollutant Emission Inventory (APEI, https://www.canada.ca/en/environment-
climate-change/services/pollutants/air-emissions-inventory-overview.html), the Co-operative
Programme for Monitoring and Evaluation of the Long-range Transmission of Air Pollutants in
Europe (EMEP, https://www.emep.int/index.html), and the MIX Asian emission inventory (Li et
al., 2017), respectively. Monthly mean aircraft emissions are generated based on the Aviation
Emissions Inventory v2.0 (Stettler et al., 2011). The aircraft particle emissions include nucleation
mode sulfate particles (Emission index $= 2\times10^{17}$ /kg-fuel, mean diameter $= 9$ nm, based on Kärcher
et al., 2000), and black carbon and primary organic carbon (POC) particles. Global biomass
burning is taken from Global Fire Emissions Database version 4 (van der Werf et al., 2017). The
volcanic emissions of $SO_2$ are taken from AeroCom point-source data (Carn et al., 2015). Fixed
global surface boundary conditions are applied for $N_2O$, CFCs, HCFCs, halons, OCS and long-
lived organic chlorine species (Eastham et al., 2014).
**2.2 Airborne ATom measurements of PNSD**
Measurements are essential in advancing our understanding of stratospheric aerosol properties
and the fundamental processes governing these properties. NASA's Atmospheric Tomography
Mission (ATom; Wofsy et al., 2021; Thompson et al., 2022) is a multi-agency effort that provides
global in situ aircraft observations of the vertical structure of aerosols from near surface to ~12 km
altitude. PNSDs are measured using the NOAA Aerosol Microphysical Properties (AMP) package
(Brock et al., 2019) comprising nucleation-mode aerosol size spectrometer(s) (NMASS)
(Williamson et al. 2018), ultra-high-sensitivity aerosol spectrometer(s) (UHSAS) (Kupc et al.
2018), and a laser aerosol spectrometer (LAS) covering aerosol sizes from 3 nm to 4.5 µm. The
aerosol number abundance can be obtained by integrating the PNSD measurements.
**2.3 The CLOUD (Cosmics Leaving OUtdoor Droplets) measurements**
Laboratory measurements of nucleation rates as a function of key controlled parameters have
been carried out in a 26.1 m³ stainless steel cylinder chamber at the European Organization for
Nuclear Research (CERN), in the framework of the CLOUD experiment (Cosmics Leaving
OUtdoor Droplets) (e.g., Kirkby et al., 2011; Kürten et al., 2016; Dunne et al., 2016). Some of
these experiments were conducted at the temperature in the range of those in the stratosphere
(Kirkby et al., 2011; Dunne et al., 2016) which are used in this study to evaluate nucleation
schemes under stratospheric conditions.
**3. Results**
**3.1 H₂SO₄–H₂O binary homogeneous nucleation (BHN) and binary ion-mediated nucleation**
**(BIMN) under stratospheric conditions**
Nucleation is one of the microphysical processes influencing particle size distributions in the
stratosphere (Turco et al., 1982) The CLOUD measurements under a wide range of well-controlled
conditions (Kirkby et al., 2011; Dunne et al., 2016) provide a unique set of data to evaluate the
nucleation theories. Yu et al. (2020) compared nucleation rates calculated based on a number of
commonly used aerosol nucleation parameterizations with the CLOUD measurements. Here we



specifically examine the comparison under stratospheric conditions where temperature is below ~
230 K. Since ammonia concentrations in the stratosphere are generally negligible, we focus on
binary nucleation in the present study.  The contribution of organics to particle formation, growth,
and compositions in the upper troposphere and LMS has been investigated in several studies (Kupc
et al., 2020; Murphy et al., 2021; Williamson et al., 2021). Because of the lack of information with
regard to the low volatile gaseous organic species, the possible role of organics in new particle
formation in LWS is not considered in the present study.

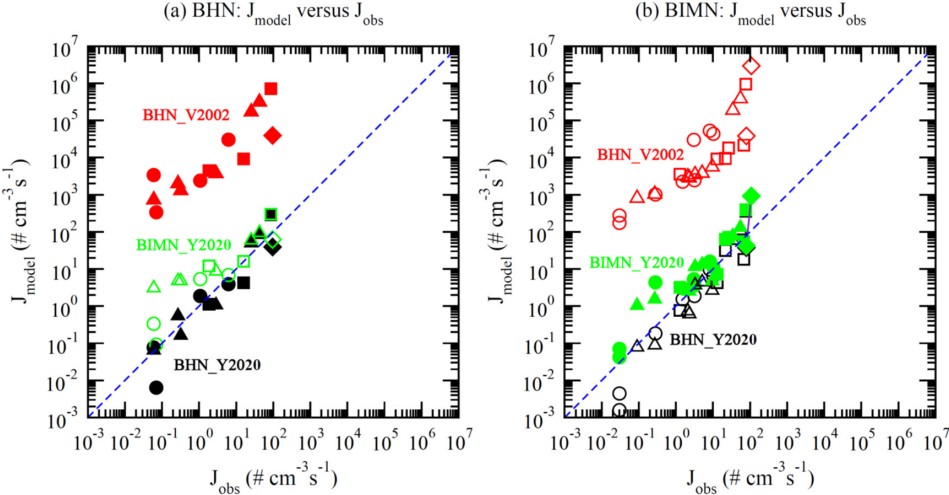

**Figure 1.** Comparison of nucleation rates based on three different schemes with CLOUD
measurements within the low temperature range (T = 205–223 K) as that in the stratosphere for (a)
binary homogeneous nucleation (no ionization) and (b) ion nucleation (at the presence of
ionization rates 2.51 – 110 ion-pairs $cm^{-3}s^{-1}$). The different nucleation schemes shown are: BHN
of Vehkamäki et al. (2002) (BHN_V2002), BHN of Yu et al. (2020) (BHN_Y2020), and BIMN
of Yu et al. (2020) (BIMN_Y2020). For comparison, under binary condition of (a), BIMN rates at
$Q = 20$ ion-pairs $cm^{-3}s^{-1}$ are given while under binary ion nucleation condition of (b), BHN rates
are also given. [$H_2SO_4$] values range from $10^6$ to $3\times10^7$ $cm^{-3}$ and are separated into four groups in
the plots (Circles: $10^6$ – $5\times10^6$ $cm^{-3}$; triangles: $5\times10^6$ – $10^7$ $cm^{-3}$; Squares: $10^7$ – $1.5\times10^7$ $cm^{-3}$;
Diamonds: $1.5\times10^7$ – $3\times10^7$ $cm^{-3}$).
204        Figure 1 compares BHN and BIMN rates based on three different schemes with CLOUD
measurements under stratospheric temperature range ($T = 205$–223 K). The nucleation schemes
shown are: BHN of Vehkamäki et al. (2002) (BHN_V2002), BHN of Yu et al. (2020)
(BHN_Y2020), and BIMN of Yu et al. (2020) (BIMN_Y2020). To show the relative importance
of homogeneous versus ion nucleation, BIMN rates at $Q = 20$ ion-pairs $cm^{-3}s^{-1}$ were given under
binary homogeneous condition in Fig. 1a and BHN rates were also given under binary ion
nucleation condition in Fig. 1b. Nucleation rates based on BHN_V2002 are consistently 3–5 orders
of magnitude higher than those observed under $H_2SO_4$–$H_2O$ binary nucleation conditions without
(Fig. 1a) and with (Fig. 1b) the effect of ionizations, while those based on BHN_Y2020 and
BIMN_Y2020 are close to the observed values. It should be noted that similar to the CLOUD



measurements with the effect of ionization, BHN rates are included in the BIMN rates (Yu et al.,
2018) and the difference between BIMN and BHN rates indicates the contribution of ion mediated
or induced nucleation. Under the conditions of Fig. 1a, assuming ionization rate of 20 ion-pairs
$cm^{-3}s^{-1}$ (within the range of its typical value in the stratosphere) the BIMN rates are about one
order of magnitude higher than BHN rates when the nucleation rates are below ~ 5 $cm^{-3}s^{-1}$ but
close to BHN rates when nucleation rates are above ~ 5 $cm^{-3}s^{-1}$. Similar difference between
BHN_Y2020 and BIMN_Y2020 can also be seen in Fig. 1b, indicating the importance of ion
nucleation at relatively lower nucleation rates (mostly associated with relatively lower [$H_2SO_4$])
and dominance of homogeneous nucleation at higher nucleation rates (associated with larger
[$H_2SO_4$]). As we show next, [$H_2SO_4$] in the background stratosphere is generally quite low and
thus ion nucleation dominates but BHN can become important in the $SO_2$ plumes injected into the
stratosphere.
**3.2 Nucleation rates and particle number concentrations in the stratosphere**
Figure 2 shows the zonal mean $SO_2$ emission (SO2_emit), particle number emitted by aviation
(PN_aviation), temperature (T), relative humidity (RH), ionization rate (Q), and [$H_2SO_4$] averaged
during the two-year period (06/2016–05/2018) covering ATom 1-4. To focus on lower stratosphere
(LS), only the values of these variables in the stratosphere (grid boxes with more than 50% time
above tropopause) are shown. The $SO_2$ emissions include all sources including volcanos and
aviation. During this period, there was one relatively strong volcanic event that injected $SO_2$ into
an altitude of ~ 14–18 km at 52°N (Fig .2a). Aviation emission is generally limited to below ~ 12.5
km altitude. Based on MERRA2 meteorology data, which is used to drive GEOS-Chem, almost
all of grid boxes at 12 km are under the tropopause in the tropics (30°N-30°S), most of grid boxes
at 12 km in the high latitude regions (60°N–90°N, 60°S–90°S) are above tropopause, and some
fractions of grid boxes at 12 km in the middle latitude regions (30°N–60°N, 30°S–60°S) are above
tropopause. As can be seen from Fig. 2b, some of aviation emissions in the middle and high latitude
regions are in the LMS, and the amount emitted into NH LMS is much higher (by several orders
of magnitude) than that in SH. The temperature in the LS ranges from 190–225K, with the lowest
value in the region just above tropical tropopause (Fig. 2c). RH in LS has highest values near
tropopause but drops quickly with increasing altitude, from ~30–50% near tropopause to ~0.1–1%
at ~ 25 km in the tropical and middle latitudes (Fig. 2d). The spatial variations of T and RH have
important effects on nucleation in LS. The cosmic ray induced ionization rate in LS has large
latitudinal gradient, ranging from ~40–100 ion-pair std. $cm^{-3}s^{-1}$ in the tropics to 100–400 ion-pair
std. $cm^{-3}s^{-1}$ in middle and high latitude region (Fig. 2e). The high ionization rates may have
important implication for particle microphysics in LS, which will also be discussed in Section 3.3.
$H_2SO_4$ is the most important aerosol precursor in LS and its concentration depends on $SO_2$
concentrations and oxidation, condensation sink, and its vapor pressure that depends on T and RH.
The annual mean [$H_2SO_4$] (Fig. 2f) has large spatial variations, ranging from a minimum of ~ 1–2
$\times 10^5$ std. $cm^{-3}$ at altitudes of ~ 12–15 km in polar regions to ~ 4–20 $\times 10^5$ std. $cm^{-3}$ close to the
tropopause. From ~ 18–25 km (well above the ATom measurement altitude), [$H_2SO_4$] increases
with altitude, mainly due to the increasing $H_2SO_4$ vapor pressure associated with vertical changes
of T (Fig. 2c) and RH (Fig. 2d).

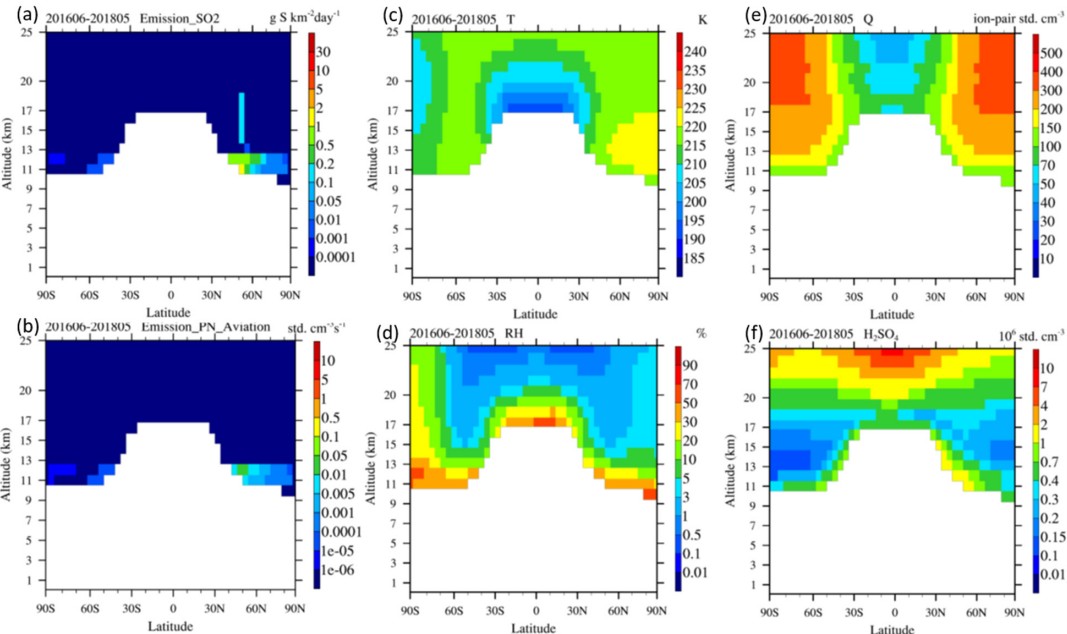

**Figure 2.** Zonal mean SO$_2$_emit, PN_Emit, $T$, RH, $Q$, and [H$_2$SO$_4$] averaged during the two-year period (06/2016-05/2018) covering ATom 1-4. To focus on the lower stratosphere, only the values of these variables in grid boxes with more than 50% time above tropopause and below 25 km are shown.

To demonstrate the effect of nucleation schemes on simulated aerosol properties, we compare in Fig. 3 zonal mean and vertical profiles of nucleation rates (J) and number concentrations of condensation nuclei larger than 3 nm (CN3) simulated based on the three nucleation schemes: BHN_V2002, BHN_Y2020, and BIMN_Y2020. In all three schemes, the aviation emissions of both SO$_2$ (Fig. 2a) and particle numbers (Fig. 2b) are the same. The model simulations indicate that NPF occurs in the lower stratosphere but is mostly confined to LMS except in the area of volcano injection (for example, above ~ 14 km around ~ 52°N). There exist large differences in the nucleation rates predicted by the three schemes (noting the logarithmic color scale), with BHN_V2002 rates generally 1–4 orders of magnitude higher while BIMN_Y2020 rates ~ one order of magnitude higher than those based on BHN-Y2020. The difference between BIMN_Y2020 and BHN_V2002 rates are smaller in the LMS over tropics (0°S-30°S) where temperature is the lowest (see Fig. 2c). The magnitudes of differences are consistent with comparisons with CLOUD measurements (Fig. 1). The difference in nucleation rates leads to substantial difference in CN3 in LMS, with those based on BHN_V2002 a factor 2–5 higher than those based on BHN_Y2020 in LMS. LMS CN3 based on BIMN_Y2020 is about 50% higher than that of BHN_Y2020. Compared to the difference in nucleation rates, the differences in CN3 is much smaller. This is expected because on one hand only a small fraction of nucleated particles survive the coagulation scavenging and grow beyond 3 nm, and on the other hand direct emission of particle numbers from aviation (Fig. 2b; treated as direct emission but most of these are actually

nucleated on chemi-ions in the exhaust plume shortly after emission) (Brock et al., 2000) and
transport provide substantial amount of CN3 even without nucleation. Nevertheless, nucleation is
still significant enough to affect the CN3. It is interesting to note that CN3 based on BIMN_Y2020
is higher at altitudes > ~ 22 km (Fig. 3h), which is associated with higher nucleation rates based
on BIMN_Y2020 than those based on BHN_V2002 and BHN_Y2020 within the altitude range of
35-55 km. It can be seen from Fig. 3 that the simulations based on three nucleation schemes all
show large hemispheric difference in particle number concentrations (by a factor of ~3-6) in LMS
at middle and high latitudes, consistent with the ATom measurements (Williamson et al., 2021).
Our sensitivity study (by turning off aviation emission, not shown, to be reported in a separate
study) indicates this large hemispheric difference is largely caused by aviation emissions,
confirming the analysis of Williamson et al. (2021).

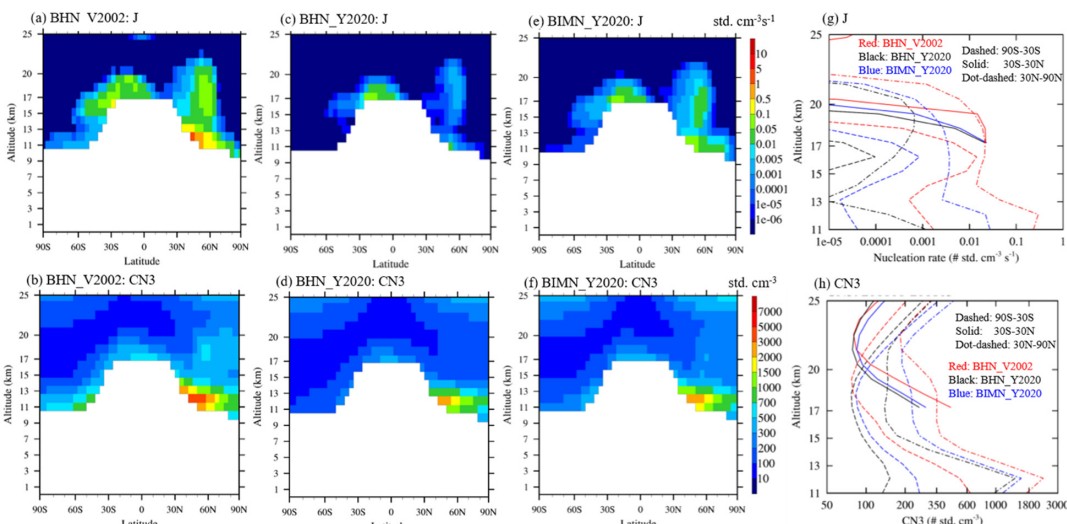

**Figure 3.** Model simulated zonal mean and vertical profiles of nucleation rates (J; upper panels)
and number concentrations of particles larger than 3 nm (CN3; lower panels) in the stratosphere
during the two-year period covering ATom 1-4 (06/2016- 05/2018), based on three nucleation
schemes (a&b: BHN_V2002, c&d: BHN_Y2020, and e&f: BIMN_Y2020). The vertical profiles
in (g) and (h) are averaged for three latitude zones (90S-30S, 30S-30N, and 30N-90N). The values
for those grids with at least 50% of time above the tropopause are shown.


303       While it is difficult to observe nucleation rates in the stratosphere, the measurement of freshly
nucleated nanoparticles can be used to constrain nucleation schemes. Figure 4a compares the
model simulated CN3 (all particles with diameter larger than 3 nm, with the upper size limit of 12
μm corresponding to the size of last model bin) based on the three nucleation schemes at altitudes
of around 12 km in SH middle and high altitudes during four seasons with the corresponding ATom
1–4 observations. As an example, Figures 4b–d show the model simulated horizontal distributions
of CN3 at 12 km altitude during ATom 4 with the values and locations of ATom4 CN3 data
overlaid.   We choose SH for comparison, as it represents the background stratosphere with



minimum influence of anthropogenic emissions (i.e., aviation) (Fig. 2b), to avoid the uncertainty
associated with aviation emissions. In Figure 4, the model results are two-month average
corresponding to the flight months of each ATom campaign while the measurement data points
shown are those sampled within the altitudes range of 11.5–12.5 km, in the stratosphere
(ozone>250 ppbv and RH<10%, following the same stratosphere definitions as in Murphy et al.
(2021) and Williamson et al. (2021)), and averaged to a 4ºx5º gridbox for comparison with
modeled results. The impact of nucleation scheme on CN3 can be clearly seen: BHN_V2002
overpredicted CN3 by a factor of 2–4, BHN_Y2020 slightly underpredicted CN3, and
BIMN_Y2020 slightly overpredicted CN3. The larger vertical spread in CN3 from BHNV_2002
is caused by the large CN3 latitude gradient associated with higher nucleation near tropopause
(Fig. 3). The comparisons above show that the ATom measurements provide a good constraint on
our understanding of the processes controlling CN3 in the LMS at mid-high latitudes.

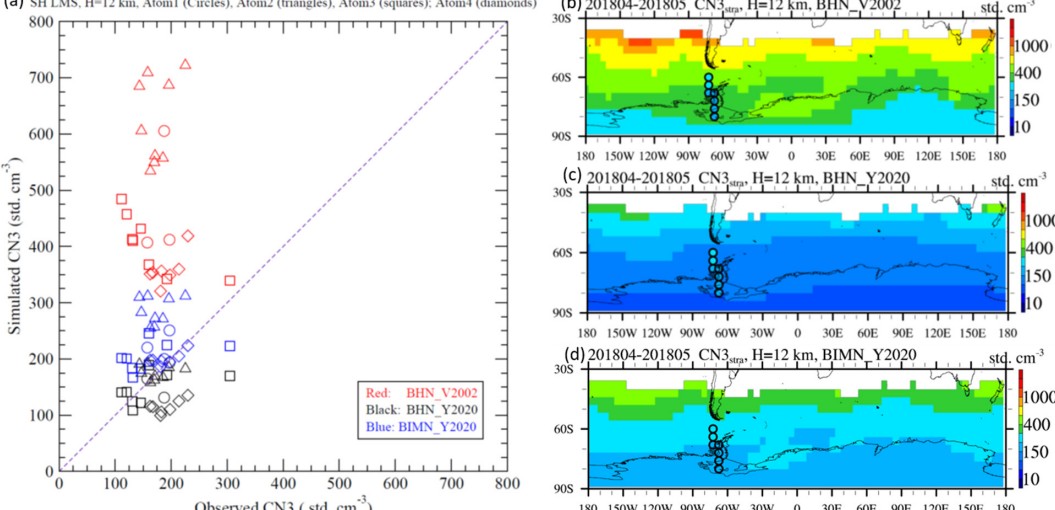

**Figure 4.** CN3 at altitudes of around 12 km in SH middle and high latitudes: (a) Model simulated
versus observed during ATom 1-4 (Circles: ATom1; Triangles: ATom2; Sqaures: ATom3;
Diamonds: ATom4); (b-d) model simulated horizontal distributions corresponding to ATom 4
based on three different nucleation schemes (BHN_V2002, BHN_Y2020, and BIMN_Y2020),
with the values and locations of ATom 4 CN3 measurements shown in the circles.

**3.3 PNSDs in the stratosphere**
Figure 5 shows the model simulated evolution of PNSDs at an altitude of 12 km over a site in
SH (70°S, 60°W) during the two-year ATom period based on the three different nucleation
schemes. The PNSDs shown in Fig. 5 are averaged into four different seasons corresponding to
the months of ATom 1-4 field campaigns and are presented in Fig. 6 for comparison with the
observed mean PNSDs in SH LMS (Williamsons et al., 2021). It should be noted that modeled
PNSDs in Fig. 6 are two-month average at one fixed site at an altitude of 12 km (in the region
where many of SH LMS measurements were taken, see Fig. 4) while the observed ones are
averaged over all SH LMS air mass sampled during the corresponding ATom campaign. While
the comparison in Fig. 6 is not exactly coterminous, it allows us to make quantitative comparisons
of modeled and observed PNSDs. To take into account the variations in both model and observed
PNSDs, standard deviations are shown as error bars in the measured and modeled curves based on
BIMN_Y2020.

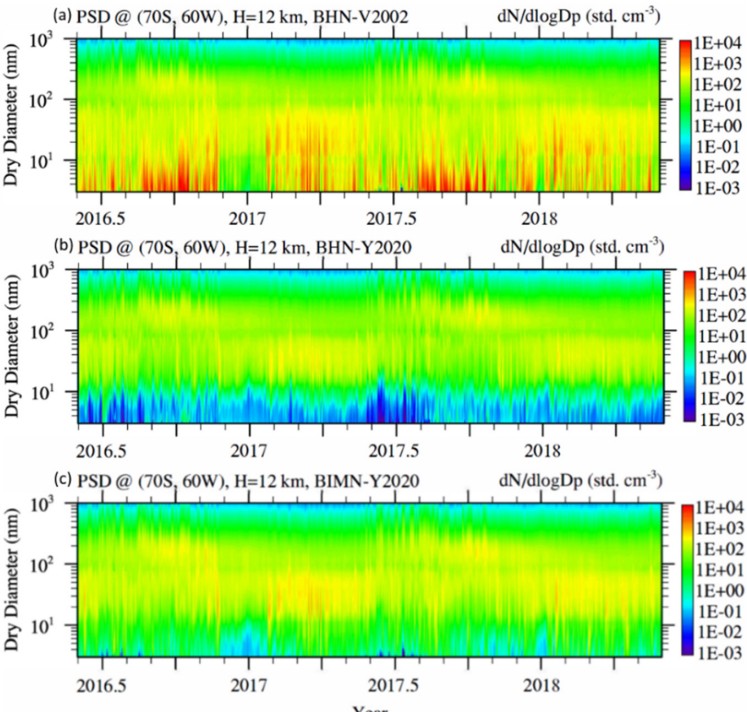

**Figure 5.** Model simulated evolution of PSDs at a site in SH (70S, 60 W) at altitude of 12 km
based on three nucleation schemes (BHN_2002, BHN_Y2020, and BIMN_Y2020).
349        Figures 5 and 6 show that PNSDs in the background LMS have multiple modes: a nucleation
mode (NuclM: ≲ 10 nm), an Aitken mode (AitkenM: ~10 − 80 nm), and two accumulation modes
(AccuM1: ~ 80 − 250 nm and AccuM2: ~ 250 − 700 nm). It should be noted that these modes
are not the same size limits as those presented in the public ATom dataset. The model based on all
three nucleation schemes generally captures the AitkenM and AccuM1 and the existence of a
minimum in PNSDs around 80 nm, although there exist differences. Interestingly, the relative
height (or peak values of dN/dlogD$_p$) of AitkenM and AccuM1 has strong seasonal variations. The
model captures a relatively higher AitkenM in SH Summer and Fall and a higher AccuM1 in SH
Spring. The model simulated PNSDs also agree well with the measurements in term of the size-
dependent standard deviations: relatively smaller standard deviations for AccuM1 and larger size
part of AitkenM and much larger standard deviations for NuclM, smaller size part of AitkenM,
and AccuM2. While the larger standard deviations for NuclM is understandable because of NPF,
it is surprising for AccuM2. The AccuM2 particles have relatively long lifetime and are expected
to be well-mixed (and thus have small variations) in LS. The transport of AccuM2 particles from
UT may contribute to the larger variations.  Murphy et al. (2021) showed the chemical signature
of this transported mode, and here we show that the variation in the size distribution may also
contain information about the mixing of UT particles into LMS. Compared to the observations,
the model simulated AccuM2 standard deviations are larger in SH Winter and Spring but are
smaller in SH Summer and Fall. The possible reasons for the large variations of AccuM2 in LMS
and the differences between model simulations and measurements remain to be studied.

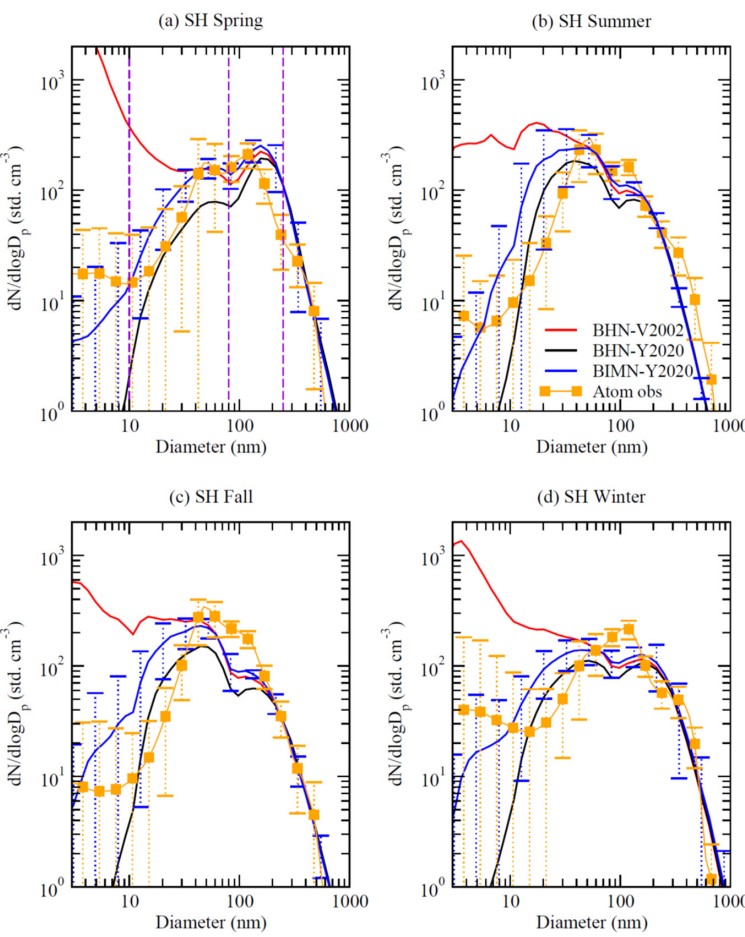

**Figure 6.** Model simulated seasonal mean PSDs at a site in SH (70ºS, 60ºW) at altitude of 12 km
based on three nucleation schemes and comparisons with the corresponding ATom measurements
(a: SH Spring 09–10/2017, b: SH Summer 01–02/2017, c: SH Fall 04–05/2018, and d: SH Winter
06–07/2016). To take into account the variations in both model and observed PNSDs, standard
deviations are shown as error bars in the measured and modeled curves based on BIMN_Y2020.
Three vertical dashed lines at 10 nm, 80 nm, and 250 nm are drawn in (a) to guide the eye to the
four modes discussed in the text.







The large impacts of nucleation schemes on PNSDs, especially those smaller than 100 nm, can
be seen in Fig. 6. The formation rates and concentrations of nucleation mode particles are very
high based on BHN_V2002 (peak dNdlogDp values reaching well above $10^3$ std. cm$^{-3}$), negligible
based on BHN_Y2020 (dNdlogDp values for particles <10 nm are generally below 1 std. cm$^{-3}$),
and moderate based on BIMN_Y2020. When compared to the observed values, the number
concentrations of particles within 3–10 nm based on BHN_V2002 are 1–2 orders of magnitude
too high but those based on BHN_Y2020 are 1–2 orders of magnitudes too low, while those based
on BIMN_Y2020 are of the same order of magnitude. The impact of nucleation schemes on NuclM
propagates into the AitkenM and AccuM1, with BHN_Y2020 giving the lowest number
concentrations while BHN_V2002 gives the highest AitkenM and BIMN_Y2020 gives the highest
AccuM1. It is interesting to note that AccuM1 based on BIMN_Y2020 is higher than that based
on BHN_V2002 although BHN_V2002 predicts higher NuclM and AitkenM, indicating a non-
linear interaction among nucleation, growth, and coagulation.
There exist a number of differences in the simulated and observed PNSDs. Firstly,
measurements indicate a slight increase of dNdlogD$_p$ with decreasing sizes for particles < 10 nm
but the simulated PNSDs based on BIMN_Y2020, the scheme mostly consistent with CLOUD
measurements and predicting NuclM concentrations closest to those observed, decreases with
decreasing sizes for particles < 10 nm. The possible reasons of the difference remain to be
investigated but probably are associated with uncertainty in nucleation rates and size-dependent
growth rates of freshly nucleated particles, and/or the fact that ATom observations are bias towards
daytime. In addition, the small number of particles in this mode is likely within the uncertainty in
the ATom measurements (about 7% of the total number of particles), so that this measured mode
may not be significant. Secondly, the model appears to overpredict the smaller size part (~10–40
nm) of AitkenM although it is close to the larger part of the mode (~40–80 nm). The overprediction
may be a result of the underestimated growth rates or coagulation scavenging rates of these
particles or overpredicted growth rates of NuclM particles. Thirdly, the model generally
overpredicts the mean mode sizes of AccuM1 and underpredicts the concentrations of the mode
except in SH Spring. The nucleation schemes have observable effects on the concentrations and
mean sizes of AccuM1 and overall the simulations based on BIMN_Y2020 are in stronger
agreement with measurements. Finally, the observed PNSDs show a clear AccuM2 in all seasons
except Fall but the model does not predict the existence of the mode at all. AccuM2 particles are
within the size range with most efficient scattering of solar radiation and thus are important for
SAI. It is therefore necessary to identify the sources of this difference and to improve the model.
As pointed out earlier, the comparison in Fig. 6 does not exactly match in terms of time and
location, which likely contributes to some of the differences shown in Fig. 6. Some of the
differences can also be caused by the uncertainties in the model in term of emissions, transport,
chemistry, aerosol microphysics, and deposition. Nevertheless, some of these differences,
especially the shape of PNSDs (AccuM2, NuclM, etc.), are unlikely to be fully accounted for by
the above-mentioned possible mismatch or model uncertainties and thus may indicate that some
fundamental processes are not represented in the model. One possible cause of the differences is
that the transport of organic-sulfate particles from UT (Murphy et al., 2014, 2021) is not properly





simulated by the model. Based on size-resolved particle composition measurements, Murphy et al.
(2021) showed that the LMS accumulation mode particles (diameter ~ 0.1 and 1.0 μm) have at
least two modes: the larger mode consists mostly of sulfuric acid particles produced in the
stratosphere, and the smaller mode consists mostly of organic-sulfate particles transported from
the troposphere. Murphy et al. (2014) showed that the fraction of organic-sulfate aerosols above
tropopause decreases quickly with altitudes. While the organic-sulfate mode aerosols from UT
may contribute to the bi-mode structure of accumulation mode particles in the LMS observed
during ATom, it is unlikely to contribute to the bi-mode structure of particles larger than ~200 nm
observed at altitude above ~20 km both in the background and in volcano perturbed stratosphere
(Deshler et al., 2013, 2019; also see Fig. 7). Here, we suggest that the role of charges on
coagulation and growth of particles in the stratosphere could be another process causing the bi-
mode of large particles in the stratosphere.
As shown Fig. 2e, ionization rates are high in LS, ranging from ~ 40–100 ion-pair std. cm$^{-3}$s$^{-1}$.
Due to their low number concentrations (~100–1000 std. cm$^{-3}$) but long lifetime, particles in the
stratosphere are expected to be in charge equilibrium. Figure 7 shows mean particle number size
distribution (PNSD) and particle volume size distribution (PVSD) observed during ATom 1-4 in
SH LMS and measured within 20-25 km altitude over Lararie WY in 1992, and fraction of particles
carrying $n$ charges based on the modified Boltzmann equilibrium equation (Clement and Harrison,
1992). The bi-mode structure of accumulation mode particles can be clearly seen in both
background and volcano perturbed stratosphere. It should be noted that while the smaller mode
generally dominates the number concentrations, the larger mode dominates mass concentrations.
Under equilibrium more particles are charged (i.e., 1-f$_0$ > 50%) than neutral (f$_0$) for particles with
diameter larger than ~80 nm and a significant fraction (> 25%) of particles larger than 300 nm
carrying multiple charges. While the equilibrium charge fraction is small for NuclM particles ($\lesssim$
10 nm), this fraction can be much larger when nucleation on ions occurs, which is consistent with
the observed overcharging of freshly nucleated particles (Laakso et al. 2007; Yu and Turco, 2008).
Particle coagulation rates are influenced by forces exerted between colliding particles, including
van der Waals and electrostatic forces, which can modify the effective collision cross section and
sticking coefficient. The van der Waals force has been shown to be important in the stratosphere
(English et al., 2011, 2012) and has been considered in the simulations shown above. The effects
of charges on coagulation and implications for PNSDs in the stratosphere have not yet been studied
(to our knowledge). Since coagulation is a dominant process for the growth of accumulation mode
particles in the stratosphere, we hypothesize that differential coagulation rates for neutral and
charged particles in accumulation modes can potentially act as a physical process separating the
modeled single accumulation mode (Fig. 6) into two modes (AccuM1 and AccuM2) as observed.
Further research is needed to test this hypothesis. In addition to affecting coagulation, charge on
small particles can also enhance the growth rate due to ion-dipole interactions of condensing
molecules with charged particles (Nadykto and Yu, 2005). This enhancement is expected to be
stronger in the stratosphere because of lower temperature (Nadykto and Yu, 2005). Beside these,
Svensmark et al. (2020) showed that the condensation of ion clusters can enhance particle growth
rates. How much the enhanced coagulation and growth rates of charged particles may shape
PNSDs and modes in the stratosphere remains to be investigated.

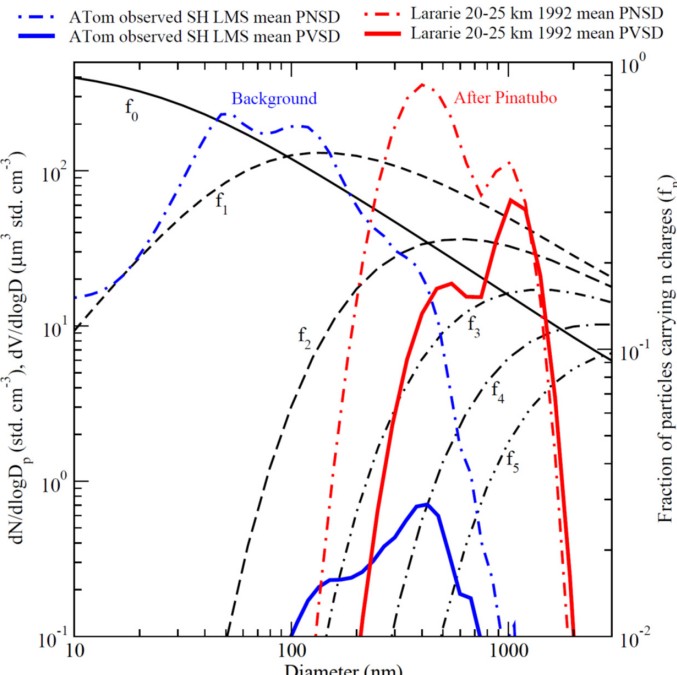

**Figure 7.** ATom 1-4 mean observed particle number size distribution (PNSD, or dN/dlogDp) and particle volume size distribution (PVSD, or dV/dlogDp) in SH LMS, balloon-borne measured mean PNSD and PVSD within 20-25 km altitude over Lararie WY in 1992, and fraction of particles carrying $n$ ($n$ = 0, 1, 2, 3, 4, and 5) charges based on the modified Boltzmann equilibrium equation (Clement and Harrison, 1992). Note that $f_n$ with $n \geq 1$ including both positive and negative charges, i.e., for example, half of $f_1$ carrying one negative charge while the other half positive.

## 4. Summary and Discussions

Interest in stratospheric aerosols has been increasing in recent years, due to the ongoing discussion about the plausibility, potential benefits and risks of offsetting climate change through stratospheric aerosol injection (SAI) to buy time for reduction of $CO_2$ in the atmosphere. Recent studies indicate the dependence of SAI efficiency on the particle size distribution (NASEM, 2021) and thus it is critical to improve foundational understanding and model representation of aerosol microphysics processes controlling the evolution of stratospheric aerosols, both under background conditions and perturbed scenarios. While formation and growth of particles in the troposphere have been extensively studied in the past two decades, very limited efforts have been devoted to understanding these in the stratosphere.

In the present study we use both CLOUD laboratory measurements taken under very low stratospheric temperatures and ATom in-situ observations of particle number size distributions (PNSD) down to 3 nm to constrain nucleation schemes and model-simulated particle size distributions in the lowermost stratosphere (LMS). We show that the binary homogenous nucleation scheme used in most of the existing SAI modeling studies overpredicts the nucleation rates by 3–4 orders of magnitude (when compared to CLOUD data), leading to significant





overprediction of particle number concentrations in the background stratosphere (by a factor of 2–4 in SH LMS, compared to ATom data). Based on a recently developed kinetic nucleation model which provides rates of both ion-mediated nucleation (IMN) and BHN at low temperatures in good agreement with CLOUD measurements, both BHN and IMN occur in the stratosphere but IMN rates are generally more than one order of magnitude higher than BHN rates and thus dominate nucleation in the background stratosphere.

In the SH LMS that has minimal influences from anthropogenic emissions, our analysis shows that ATom-measured PNSDs generally have four apparent modes: a nucleation mode (NuclM: ≲ 10 nm), which may not be statistically significant, an Aitken mode (AitkenM: ~10–80 nm), and two accumulation modes (AccuM1: ~ 80–250 nm and AccuM2: ~ 250–700 nm). The model generally captures the AitkenM and AccuM1 and the existence of a minimum in PNSDs at ~ 80 nm, although there are differences. The model captures a relatively higher AitkenM in SH Summer and Fall and a higher AccuM1 in SH Spring. The model simulated PNSDs also agree well with the measurements in term of the size-dependent standard deviations: relatively smaller standard deviations for AccuM1 and larger size part of AitkenM and much larger standard deviations for NuclM, smaller size part of AitkenM, and AccuM2.

A detailed comparison indicates the existence of a third PNSD mode peaking around 300–400 nm in the ATom measurements that are not captured by the model. Compared to the observations, the model-simulated AccuM2 standard deviations are larger in SH Winter and Spring but are smaller in SH Summer and Fall. In addition, the model overpredicts the number concentration of particles in the size range of 10–50 nm. These differences may indicate that, in addition to nucleation, the model may be missing some fundamental microphysical processes of stratospheric aerosols. Our analysis shows that, in the stratosphere, more particles are charged (positive + negative) than neutral for particles with diameter larger than ~80 nm and a significant fraction (> 25%) of particles larger than 300 nm carrying multiple charges. We propose that the role of charges on coagulation and growth of particles in the stratosphere, where ionization rates are high and particles have very long lifetime, is likely one of such processes. Considering the importance of accurate particle size distributions (especially the accumulation mode particles) for projecting realistic radiative forcing response to stratospheric aerosols, it is essential to understand and incorporate such potentially important processes in model simulations of future changes in the stratosphere.

**Conflict of interest**: The authors declare that they have no conflict of interest.

**Acknowledgments.** The MERRA-2 data used in this study have been provided by the Global Modeling and Assimilation Office (GMAO) at NASA Goddard Space Flight Center. This research has been supported by NASA (grant nos. 80NSSC19K1275 and 80NSSC21K1199) and SilverLining.

**Data availability.** The GEOS-Chem model is available to the public at https://geos-chem.seas.harvard.edu/. Simulation output in this analysis is available at https://doi.org/10.5281/zenodo.6909944. The ATom dataset is published as Wofsy et al., (2021, https://doi.org/10.3334/ORNLDAAC/1925) and is also available at https://espoarchive.nasa.gov/archive/browse/atom (last access: June 2022).



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
