# Peer review of "Particle number concentrations and size distributions in the stratosphere: Implications of"

_Atmospheric Chemistry and Physics, 2022_

## Author Comment (AC1)

- **RC1**: 'Comment on acp-2022-487', Anonymous Referee #1, 26 Aug 2022

The authors thanks Referee #1 for the constructive comments, which have helped us to clarify and improve the manuscript. Below we address the comments, with the reviewer comments in black, and our response in blue. We have revised the manuscript accordingly. All changes made to the manuscript have been marked in the submitted Track-Changes version.

Manuscript by Fangqun et al studies nucleation in the stratospheric condition by evaluating different nucleation schemes against CLOUD laboratory measurements and ATom observations in the southern hemisphere stratosphere. Study showed that in the recently developed kinetic nucleation model nucleation rate is much better in agreement with measurements compared to widely used binary homogeneous nucleation. This paper is exactly the type of research that modelers of stratospheric aerosols need, especially if you are studying stratospheric aerosol injections. Good indicator for excellent study is that after reading the paper, you want to take something from it to your own research. In this case it is the nucleation scheme.

Manuscript is well written, and when questions came to my mind, those were answered in the following lines. Or at least as well as possible. Overall this is an interesting and excellent study and it is difficult to find any major or even minor issues. I have just few very minor comments which would clarify some specific points:

We appreciate the positive comments and confirmation of the importance of this work. Please see below for our point-to-point replies and clarifications.

Model and data section:

It would be interesting to know how competition between nucleation and condensation for sulfuric acid vapor is done in  GC-APM.

In GC-APM, nucleation is calculated before condensation using a time-splitting technique. Therefore, no competition between nucleation and condensation for sulfuric acid vapor is considered. In most conditions, nucleation consumes only a very small fraction (<1%) of sulfuric acid vapor in the air and the time splitting does not affect the results.  When nucleation rate is high, reduced time step for nucleation and growth is used to ensure that the fraction of sulfuric acid vapor consumed by nucleation each time step is small. The GC-APM uses a semi-implicit scheme to calculate sulfuric acid condensation together with sulfuric acid gas phase production to ensure that the change of sulfuric acid vapor concentration is smooth. We have clarified this in the revised text.

P6 L206, After reading the abstract and introduction I got the impression that in this paper only the BHN of Vehkamäki and new BIHM scheme are evaluated. This line is the first time where BHN of Yu et al. (2020) is mentioned. I think it would be good to somehow introduce this nucleation scheme earlier (just by 1 or few line(s)) and say with just few words how it differs from the BHN of Vehkamäki (new look up tables(?) I assume).

Modified as suggested:

"Figure 1 compares nucleation rates based on the following three different schemes with CLOUD measurements under stratospheric temperature range ($T$ = 205–223 K): BHN of Vehkamäki et al. (2002) (BHN_V2002), BHN of Yu et al. (2020) (BHN_Y2020), and BIMN of Yu et al. (2020) (BIMN_Y2020). BHN_V2002 and BHN_Y2020 differ in term of thermodynamic data and nucleation approach used (Yu et al., 2020)."

P6 Figure 1 (and later figures): Really minor thing but it would be more clear if (a) and (b) were upper left of the panel and not in the title and that order would be a-b-c in upper panels and not a-c-e as in figs 2-3.

Modified as suggested.

P7 L246 and later in the text, at least I am not familiar with "std" in units. Could this be opened up?

Here "std. cm$^{-3}$" refers to per cubic centimeter at standard temperature and pressure, 273 K and 1013 hPa, respectively. This has now been specified when ""std. cm$^{-3}$" is firstly used.

P7 L251 and L253, brackets in [H2SO4]

Brackets in [$H_2SO_4$] refer to $H_2SO_4$ vapor concentrations. This is now pointed out when [$H_2SO_4$] is first used.

P11 L349 You could clarify that here you refer to measurements even though it is obvious after reading the next couple of lines.

Modified as suggested.

P12 L361-368 I have been struggling with this and I am not sure if the standard deviation in the largest model is surprisingly large or is it even large or not. Number concentration in AccuM2 is much lower compared to AccuM1 and Aitken modes and thus in the linear scale the size range of the standard deviation bar is not actually large as it seems to be in logarithmic scale.

You are correct -- the absolute values of the standard deviation in AccuM2 are actually smaller than those in AccuM1 and Aitken modes. What we meant in the discussion is the normalized standard deviation (i.e., the standard deviation divided by the mean) that shows the relative variations. We have clarified this in the revised text.

Figures 5 and 6 and captions: There abbreviation of particle number size distribution is "PSD" while in the text it is "PNSD"

Good catch. All "PSD" have been changed to "PNSD" for consistency.

P13 L389-391 I think this is kind of expected as there is lower nucleation rate in BIMN compared to BHN_V2002, there is more available sulfuric acid vapor for condensation. However, the BHN-V2020 line is lower than BIMN and BHN_V2002 regardless the size of

the aerosol, which is interesting. It means that particulate sulfate (+small amounts of some other species) burden is lower in BHN-V2020 compared to others. Where is this "missing" sulfate in BHN-Y2020? In the gas phase, or in some other location, or removed from the atmosphere? If you could easily give an answer to this, it would be interesting to know.

Actually, there is no "missing" sulfate in BHN-Y2020. While the line of BHN-Y2020 is lower than that of BIMN and BHN-V2002 for particles of smaller sizes (<~ 300 nm), it is slightly higher for larger particles (>~ 300 nm). In PNSD plots, these slightly higher values for BHN-Y2020 for larger particles cannot be clearly seen in the PNSD plots but can be seen in the values given in the data files or mass size distributions. This is consistent with the competition of sulfuric acid gas between pre-existing larger particles and nucleated smaller particles. This has been clarified in the revised text.

P13 L408 I find this bi-mode structure of accumulation size region really interesting. What do you think, should this be taken into account in modal aerosol schemes especially in stratospheric conditions and add one extra mode between usually used accumulation and coarse mode? Your results are not the only case which would speak for it. Of course then in the modal scheme bi-mode structure would be artificial, as it still remains unknown what is causing it.

More research is needed to characterize and understand this bi-mode structure. If it is indeed a common and wide-spread feature, it should be taken into account both in modal and sectional aerosol schemes. We agree that it is critical to understand what is causing it.

P14 L439 Is there bi-mode structure of accumulation mode in a volcano perturbed atmosphere? I would say that there is the coarse mode and one accumulation mode.

It depends on how we define coarse mode sizes. In the aerosol research community, coarse mode refers to particles generally larger than 2.5 µm (i.e, PM2.5, PM10, etc.). In this case, we can still say that there is a bi-mode structure of accumulation mode in a volcano perturbed atmosphere.

---

## Author Comment (AC2)

- **RC2**: 'Comment on acp-2022-487', Anonymous Referee #2, 28 Aug 2022

The authors thanks Referee #2 for the thoughtful comments, which have helped us to clarify and improve the manuscript. Below we address the comments, with the reviewer comments in black, and our response in blue. We have revised the manuscript accordingly. All changes made to the manuscript have been marked in the submitted Track-Changes version.

This is really interesting. I like the combination of modeling, field experiment measurements, and chamber measurements. Getting a chamber to cooperate with stratospheric conditions is no small feat. There are some important gaps in the study that I'd like to see resolved, mainly having to do with the applicability of your datasets.

More specifically, you show that these new nucleation schemes better match observations. But your observations do not match what is usually thought of as hypothetical SAI conditions. This introduces a potential source of error in your study that is not well discussed.

We appreciate the positive comments about this work. Please see below for our point-to-point replies and clarifications about the gaps.

Comments:

I'd like to see you discuss volcanic eruptions more. Presumably if you're coming up with new assessments of past modeling of SAI it would also affect past modeling of volcanoes. Did we miss something very important in our previous assessments of volcanic aerosol microphysics? Did that affect our estimates of radiative forcing or chemistry?

Detailed measurements are needed to properly assess the modeling performance. This study focuses on the period where in-situ ATom airborne measurements are available. As the reviewer pointed out in the next comment, the ATom measurement period does not have a high stratospheric loading. It remains to be investigated if previous assessments of volcanic aerosol microphysics missed something important. We expect the uncertainties in the nucleation schemes and unknown cause of the bi-modal structure of accumulation mode particles will affect particle optical properties and surface area and thus radiative forcing or chemistry. The exact effects remain to be studied, ideally with good in-situ particle size distribution measurements such as those from ATom. We have added some discussions on this in Section 4.

The period chosen (which overlaps with ATom) doesn't have a high stratospheric loading, and the particle size is substantially smaller than would be experienced under SAI. Is there any reason to think that microphysical behavior will be different under SAI conditions (or volcanic conditions)? This is exemplified in Figure 4 – while it's clear that the updated schemes better match observed CN3 than the 2002 scheme, this is only for a narrow range of CN3 and is poorly constrained for higher CN3 numbers.

We agree that the particle size during the ATom period is substantially smaller than would be experienced under SAI. The stratospheric particle properties during the ATom period can be considered to be those of background stratosphere. We expect that microphysical behavior

described in this work will be similar under SAI conditions (or volcanic conditions). However, the effect under these conditions remains to be investigated. We agree that Figure 4, representing the background LMS, is only for a narrow range of CN3. Unfortunately, we do not have measurements for higher CN3 numbers that can be used to constrain the model.

12 km isn't very high in altitude – that won't reach the stratosphere in many places, so the fact that your scheme better matches observations doesn't necessarily show that it better matches observations in the stratosphere. I would like to see more discussion on how this limitation affects your conclusions about stratospheric NPF. You discuss some of this in Section 3.2, but I'm having trouble interpreting the applicability and limitations of your study. Relatedly, on lines 233-234, which volcanic event and how much SO2?

Yes, this is a limitation of current measurements – ATom can only reach up to ~ 12 km altitude. Similar measurements (i.e., particle size distributions down to ~ 3 nm) at higher altitudes will be needed to evaluate nucleation schemes at higher altitudes. It should be noted that, in addition to using ATom data, in this work we also use the CLOUD laboratory measurements to assess the nucleation schemes (Fig. 1).

The volcano mentioned on lines 233-234 is the eruption of the Bezymianny volcano (55.98˚N, 160.59˚E) on December 20, 2017. The amount of $SO_2$ injected to the stratosphere (up to 18 km) due to this volcanic was $5 \times 10^6$ kg S according to the volcano emission inventory (Carn et al., 2015). We have added the information of this volcano in the revised text.

Carn, S. A., Yang, K., Prata, A. J. and Krotkov, N. A.: Extending the long-term record of volcanic SO2 emissions with the Ozone Mapping and Profiler Suite nadir mapper. Geophys. Res. Lett., 42: 925– 932. doi: 10.1002/2014GL062437, 2015.

I'd like to see more description about the chamber. There is more to the stratosphere than just cold temperature – one needs to include low pressure, harsh radiation, composition, etc. Are you actually reproducing stratospheric conditions or just stratospheric temperatures? And if the latter, how relevant are your conclusions for stratospheric NPF?

Just the stratospheric temperatures. NPF in the stratosphere is generally considered to be involving $H_2SO_4$-$H_2O$. The physics underlying the $H_2SO_4$-$H_2O$ nucleation is generally well understood although there is uncertainty in the thermodynamic data of pre-nucleation clusters. Presently we do not have theoretical and experimental evidence indicating the effect of low pressure and harsh radiation. The effect of $H_2SO_4$-$H_2O$ binary composition is taken into account by the present nucleation schemes.

Figures 2, 3, and 5: I don't have a good sense for which scheme gives you better answers. What are these "supposed to" look like?

These figures show us the large difference caused by different nucleation schemes. Unfortunately, we do not have measurements to tell us what these are "supposed to" look like. Nevertheless, measurements given in Figures 1, 4, and 6, although limited, did provide some constrains on what these are supposed to look like under the conditions specified.

You make a good case for a second accumulation mode.  But there are many schemes (both modal and sectional) that take a second accumulation mode into account.  Perhaps they don't get the processes correct that would create such a mode, but they do have it.  It might be useful to point out what those schemes are doing wrong.

We were not able to locate specific references showing "many schemes (both modal and sectional) that take a second accumulation mode into account". While some modal schemes use different modes (like MAM) to represent particles of different sources and sizes, we did not find any model to specifically separate accumulation mode particles in the stratosphere into two modes.

You could do a bit more work (or some discussion) to characterize your uncertainty.  On lines 392-431 you discuss several sources of potential error, including missing processes or uncertainty in nucleation rates.  Do you have a sense as to whether these sources are dominant or secondary?  If the former, your results are at the risk of being made obsolete by someone who addresses those other sources of error.

It is hard to robustly quantify various uncertainties for the reasons pointed out in the main text. A certain source could be dominant in certain aspect (for example, the cause of the second accumulation mode) while others could be secondary. We raised these issues in the study so that the research community are aware of these. We will be happy to see these sources of errors being addressed in future research toward advancing our understanding of processes controlling size distributions of stratospheric aerosols.

---

## Author Comment (AC3)

- **RC3**: ['Comment on acp-2022-487'](), Anonymous Referee #3, 13 Sep 2022

The authors thanks Referee #3 for the constructive comments, which have helped us to clarify and improve the manuscript. Below we address the comments, with the reviewer comments in black, and our response in blue. We have revised the manuscript accordingly. All changes made to the manuscript have been marked in the submitted Track-Changes version.

Very good work! The paper compares the nucleation process, between model results with two different nucleation schemes and CLOUD/ATom measurements, under the stratospheric condition. It's nice to see the authors relate the study to the SAI simulation, which could help to improve the SAI modeling accuracy. The overall reasoning in the paper is solid and well-justified.

We appreciate the positive comments and confirmation of the solid quality of this work.

I have some minor comments and corrections:

Line 81-83: Providing only two publication examples (i.e., Weisenstein et al., 2022, Laakso et al., 2022) seems not enough to prove that "BHN_V2002 has been used in most SAI modeling studies". It would be more convincing if the authors can tell us how many models use BHN_V2002. For example, there are many models involved in GeoMIP (Kravitz et al., 2013), it would be helpful if the authors can tell the GeoMIP community how many GeoMIP simulations use BHN_V2002 for nucleation simulation.

The two cited references are recent SAI model intercomparison papers. According to Kravitz et al. (2013), 13 out of 16 Geo-MIP models assumed prescribed or bulk stratospheric aerosol. The remaining 3 models generalized aerosols from $SO_2$ but no information on the nucleation schemes used was given in the paper. We have checked more SAI-related papers that explicitly consider size-resolved particle microphysics (including nucleation) and have revised the text regarding the BHN_V2002 scheme used in these studies.

 "Indeed, the $H_2SO_4$–$H_2O$ binary homogenous nucleation (BHN) parameterization developed two decades ago by Vehkamäki et al. (2002) (named BHN_V2002 thereafter) has been widely used in SAI modeling studies when nucleation process is explicitly considered (e.g., Tilmes et al., 2015; Jones et al., 2021; Weisenstein et al., 2022). Tilmes et al. (2015) described a Geoengineering Model Intercomparison Project (GeoMIP) experiment designed for climate and chemistry models, using the stratospheric aerosol distribution derived from the ECHAM5-HAM microphysical model (Stier et al., 2005) which calculated nucleation rates with the BHN_V2002 scheme. Both models (UKESM1 and CESM2-WACCM6) employed for a recent GeoMIP G6sulfur study (Jones et al., 2021) used the BHN_V2002 scheme. In another recent SAI study based on three interactive stratospheric aerosol microphysics models (Weisenstein et al., 2022), two models (MAECHAM5-HAM and SOCOL-AER) used BHN_V2002 scheme while the other (CESM2-WACCM) used an empirical nucleation scheme to calculate nucleation rate as a function of sulfuric acid concentration only (i.e, no dependence on temperature and relative humidity)."

Jones, A., Haywood, J. M., Jones, A. C., Tilmes, S., Kravitz, B., and Robock, A.: North Atlantic
    Oscillation response in GeoMIP experiments G6solar and G6sulfur: why detailed modelling is needed
    for understanding regional implications of solar radiation management, Atmos. Chem. Phys., 21,
    1287–1304, https://doi.org/10.5194/acp-21-1287-2021, 2021.
Stier, P., Feichter, J., Kinne, S., Kloster, S., Vignati, E., Wilson, J., Ganzeveld, L., Tegen, I., Werner, M.,
    Balkanski, Y., Schulz, M., Boucher, O., Minikin, A., and Petzold, A.: The aerosol-climate model
    ECHAM5-HAM, Atmos. Chem. Phys., 5, 1125–1156, https://doi.org/10.5194/acp-5-1125-2005,
    2005.
Tilmes, S., Mills, M. J., Niemeier, U., Schmidt, H., Robock, A., Kravitz, B., Lamarque, J.-F., Pitari, G.,
    and English, J. M.: A new Geoengineering Model Intercomparison Project (GeoMIP) experiment
    designed for climate and chemistry models, Geosci. Model Dev., 8, 43–49,
    https://doi.org/10.5194/gmd-8-43-2015, 2015.

Line 136: 4°x5° is too coarse. If possible, please repeat the simulations in 2°x2.5°. If not, the authors should discuss how much the grid resolution may influence the difference between the model results and observations, especially for the comparison between model results and ATom observations at one site in Figure 6.

We chose 4°x5° in this study because of computing cost constrain. It took about 2.5 days of wall clock time for one-year 4°x5° simulation and about 20 days of wall clock time for one-year 2°x2.5° simulation (more # of grid boxes plus shortened time step). Due to relative long lifetime of stratospheric aerosols, at least 1.5 years of spin-up time is generally required.

[Figure]

Figure R1. Same as Figure 6a in the main text but with a curve (cyan) added for BIMN-Y2020 case simulation at 2°x2.5° horizontal resolution.

While 4°x5° resolution is not ideal, it can be justified for the study of relatively homogeneous horizontal distributions of aerosols in the stratosphere. It should be noted that for comparisons shown in Fig. 6, the ATom data has been averaged to a 4°x5° gridbox (see Fig. 4). To check the effect of grid resolution, we run the BHN_Y2020 case at 2°x2.5° for two years (2016-2017). With the first 1.5 years as spin-up, we compare the modeling results with ATom 3 (09-10/2017, SH Spring) in Figure R1 (corresponding to Fig. 6a). It can be seen that the difference is generally small compared to the variations of the measurements and model simulations (as indicated by the error bars) and thus will not affect the main conclusions of this paper. Please note that at least some of the difference between the blue (4°x5°) and cyan (2°x2.5°) curves is caused by the different areas represented by the two curves (one for a 4°x5° grid box and the other for a 2°x2.5° grid box).

Line 274: why the tropics are selected as "(0°S-30°S)" instead of "(30°S-30°N)"?

Thank you for pointing this out. It was a typo. Corrected.

Line 296: Figure 3 needs to be optimized:

(1) Set shared x or y axis label among figures (a) to (f).

y-axis label is now shared. We keep x-axis label for each panel for clarity.

(2) Adjust the location/size of figures (g) and (h).

Slightly adjusted.

(3) There is a horizontal dashed line on the top of the figure (h), which should be deleted.

Fixed.

Line 296: for Figure 3 (g),

(1) is it a coincidence that three solid lines end up with a similar nucleation rate (about 0.02 std. cm$^{-3}$ s$^{-1}$) at approximately 17.5 km?

This is a coincidence for 30S-30N average. As can be seen from Fig. 3a-3c at ~17.5 km, there exist variations in the latitude direction although the 30S-30N average is about the same.

(2) why there is an elbow point (at around 20 km) in the red solid line? In another word, why does the nucleation rate from BHN_V2002 has a much larger changing rate with height above 20 km, compared to below 20 km?

This is a good observation. It is caused by much smaller vertical gradient in BHN_V2002 nucleation rates within ~ 17-20 km (see Fig. 3a), likely a result of different dependences of

nucleation rates based on different schemes on $T$, RH, and [$H_2SO_4$] which have large vertical variations (see Fig. 2). We have pointed this out in the revised text.

Line 319: "BHNV_2002" should be "BHN_V2002".

Corrected.

Line 390-391: I think that the competition between nucleation and condensation mentioned by Laakso et al. (2022) might be a complement to the "nonlinear process" (Line 390-391) mentioned by the authors.

Yes. We added a sentence to point out the work of Laakso et al. (2022) with regard to the competition: "The competition between nucleation and condensation for available sulfuric acid gas has been shown to be important for SAI studies (Laakso et al., 2022)."

Line 409: I don't understand the sentence: "Finally, the observed PNSDs show a clear AccuM2 in all seasons except Fall but the model does not predict the existence of the mode at all."

Based on Figure 6, the model may underpredict the AccuM2, especially in summer. But we cannot say "the model does not predict the existence of the AccuM2 mode at all".

What's more, the authors say "the model-simulated AccuM2 standard deviations are larger in SH Winter and Spring but are smaller in SH Summer and Fall" in Line 505. If "the model does not predict the existence of the AccuM2 mode at all", there would be no "model-simulated AccuM2 standard deviations".

That's a good and valid point. "underpredict" is a more accurate word for this. We have modified the sentence to reflect this.

"Finally, the observed PNSDs show a clear AccuM2 in all seasons except Fall but such a mode cannot be clearly seen in the model simulated PNSDs, indicating that the model underpredicts the concentrations of AccuM2 mode particles."

Line 437: The citation (Clement and Harrison, 1992) is missed in the References. Please check and make sure all the citations in the main text are correspondingly listed in the References.

Thanks for noticing this. We have added the references and double-checked other citations.

Line 475: Suggest changing "SAI efficiency" to "SAI radiative efficacy". Radiative efficacy refers to the radiative forcing normalized by the aerosol injection rate, which is widely used in SAI studies (e.g., Dai et al., 2018).

Modified as suggested: "Recent studies indicate the dependence of SAI radiative efficacy (Dai et al., 2018) onthe particle size distribution (NASEM, 2021) …"

In the discussion part, I think the authors can highlight the importance of model development for reducing model uncertainties of SAI simulations. Some other SAI-related model development work (e.g., Golja et al., 2021, Sun et al., 2022) is worth mentioning.

Yes, we have added the following sentence in the discussion part: "The present work highlights the importance of advancing scientific understanding of processes controlling properties of stratospheric particles as well as further development, improvement, and validation of models for reducing uncertainties of SAI simulations (e.g., Golja et al., 2021, Sun et al., 2022)."

For the next step, I hope the authors could consider comparing the modeled aerosol radiative forcing based on the two different nucleation schemes, which could help the Solar Geoengineering community to have a clear feeling about how much can different nucleation schemes influence the SAI radiative efficacy.

Thanks for the suggestion. That's our plan for the near future.

The papers mentioned above:

Dai et al., 2018: https://doi.org/10.1002/2017GL076472

Golja et al., 2021: https://doi.org/10.1029/2020JD033438

Kravitz et al., 2013: https://doi.org/10.1002/2013JD020569

Laakso et al., 2022: https://doi.org/10.5194/acp-22-93-2022

Sun et al., 2022: https://doi.org/10.1029/2021MS002816

---

## Author Response (AR2)

Dear Andreas,

Thank you very much for obtaining the review comments of our manuscript and your suggestion. Below is my reply to the reviewer's comments and your suggestion (in blue).

The reviewer responded to me with the following comment:
"The authors have done a good job of pointing out that the questions I asked are really hard and probably can't be answered in one paper. But that also means that their paper probably shouldn't be so conclusive. I'd like to see them take more of the tack that they mention in their reply - pointing out shortcomings that the research community needs to pay attention to in the coming years. I think a paper along the lines of "There's a lot we don't know, and it's really important for us to find out" would be more useful than a paper that limits itself to narrow descriptions of the results of their tools. I think this is more of a comment about framing than anything - the authors don't need to come up with new results to address my comments."

To me, it seems that this comment can be taken up in the summary and discussion section.
Please let me know how you want to approach the comment by the reviewer.

I agree with the reviewer about the importance of "*pointing out the shortcomings that the research community needs to pay attention to in the coming years*". Actually this is the main focus of this manuscript by pointing out (and showing) the limitation of the nucleation scheme (i.e., BHN_V2002) widely used in the community and the unknown cause of the bi-modal structure of accumulation mode particles. I also agree that "*There's a lot we don't know, and it's really important for us to find ou*t". To take your suggestion, I have added a sentence in the Summary and Discussion section to emphasize these points. I have also slightly modified the last sentence of the abstract to reflect this. The changes are marked in red below.

Summary and Discussion
… It remains to be investigated if previous assessments of volcanic aerosol microphysics missed something important. We expect the uncertainties in the nucleation schemes and unknown cause of the bi-modal structure of accumulation mode particles will affect particle optical properties and surface area and thus radiative forcing or chemistry. In addition to what we have shown in this study, there are likely other uncertainties or missing processes we do not know and the community needs to identify and resolve these. The present work highlights the importance of advancing scientific understanding of processes controlling properties of stratospheric particles, identifying important processes that the present models might have missed, and further development, improvement, and validation of models for reducing uncertainties of SAI simulations (e.g., Golja et al., 2021, Sun et al., 2022).

Abstract:
… Considering the importance of accurate PNSDs for projecting realistic radiation forcing response to stratospheric aerosol injection (SAI), it is essential to understand and incorporate such potentially important processes in SAI model simulations and carry out further research to find out what other processes that the present models might have missed.

Please let me know if you have any questions.

Best regards,

Fangqun